# Why Organizational Commitment and Work Values of Veterans Home Caregivers Affect Retention Intentions: A Social Exchange Theory Perspective

**DOI:** 10.3390/healthcare13192396

**Published:** 2025-09-23

**Authors:** Szu-Han Yeh, Kuo-Chung Huang

**Affiliations:** Department of Business Administration, Nanhua University, Chiayi County 622301, Taiwan; kchuang@nhu.edu.tw

**Keywords:** interpersonal relations, personnel turnover, social values, employee participation

## Abstract

**Background/Objectives:** The stability of caregiver manpower plays a crucial role in the operation of long-term care institutions. This study adopts Social Exchange Theory as the theoretical foundation to construct the psychological mechanism through which organizational commitment and work value influence retention intention via job involvement. Against the backdrop of Taiwan’s intensifying aging society and the increasing service demands of the veterans’ support system, Veterans Homes have gradually become indispensable within the long-term care system. Therefore, the primary objective of this study is to explore the formation mechanism of retention intention among caregivers in Veterans Homes. **Methods:** Data analysis was conducted using structural equation modeling, with 447 valid samples collected from caregivers across 16 Veterans Homes in Taiwan. **Results:** The results indicate that, in the process of forming retention intention, job involvement serves as a mediator between organizational commitment and work value on retention intention and demonstrates significant mediating effects. **Conclusions:** These findings suggest that when caregivers perceive value realization and organizational identification in their work, they are more likely to exhibit active engagement, thereby strengthening their tendency to remain employed. Furthermore, the study reveals that the effect of organizational commitment on job involvement is stronger than that of work value, indicating that exchange motives triggered by emotional bonds carry greater implications for retention. In conclusion, organizational support and personal value perceptions stimulate emotional engagement, which further influences caregivers’ decisions to remain in long-term service and ultimately shape their retention behavior.

## 1. Introduction

Against the backdrop of an increasingly aging society in Taiwan and the growing demand for services within the veterans’ support system, Veterans Homes have gradually assumed an indispensable role in the long-term care system. Since their establishment in 1953, Veterans Homes have served as important institutions providing care services for elderly retired military personnel, particularly those without dependents or with limited financial resources. At present, there are 16 Veterans Homes across Taiwan. From May 2022 to May 2023, the monthly average number of resident veterans ranged between approximately 5337 and 5489 [1], creating a substantial care demand system. Within this structure, the caregiving responsibilities undertaken by caregivers directly affect the overall quality of care and the well-being of residents; however, workplace conditions and human resource issues have gradually attracted increasing attention.

In fact, caregivers experience demanding workloads with medication counseling, care management, blood pressure monitoring, blood glucose monitoring, etc. [2]. This leads to high levels of work stress, difficulty in replacing labor, and a persistently high turnover rate [3]. And the shift in career preference among the millennial labor often reduces retention intention. Landrum [4] reported that over 40 percent of young graduates may leave their original organization within two years of employment, and more than 70 percent will experience multiple job changes before the age of 35. Turnover not only imposes cost burdens related to recruitment, training, and workforce replenishment but also risks the loss of tacit organizational knowledge, potentially affecting customer satisfaction and undermining internal morale stability [5]. Therefore, reducing the risk of voluntary turnover among caregivers and promoting stable retention has become a critical issue for the sustainable development of organizations.

Organizational commitment has long been regarded as one of the key psychological antecedents of stable employee retention [6,7]. Given its close relationship with job satisfaction, organizational commitment fosters a professional atmosphere and supports employees’ career development [8], thereby facilitating the continued development of work identification and a sense of responsibility in highly demanding work contexts [9]. Moreover, research in the field of organizational behavior has consistently identified low organizational commitment as an important antecedent of voluntary turnover [7,10]. Thus, this study considers organizational commitment to be a variable of significant relevance. In recent years, work value has received growing attention in relation to retention intention [11]. Work value reflects employees’ intrinsic beliefs and perceptions of meaning in the workplace [12,13]. Different work values guide employees’ behavioral choices and help shape their work motivation and level of engagement [14]. Accordingly, this study incorporates work value into the analysis to better understand how employees assess work situations based on personal values and decide whether to remain in their positions.

In addition, job involvement has been recognized in prior research as a critical factor in organizational behavior [15,16]. In the present study, job involvement is conceptualized as a psychological process that not only promotes employee performance and satisfaction [17] but may also serve as a motivational pathway through which organizational commitment and work value influence retention intention. However, past research has seldom explored this aspect. The existing literature on caregiver retention intention has primarily focused on medical institutions and general long-term care facilities [18,19], while research on Veterans Homes, which are characterized by a unique system and service recipients, remains relatively scarce.

Therefore, this study targets caregivers in Veterans Homes with two primary objectives: (1) to examine the direct effects of organizational commitment and work value on retention intention, and (2) to assess the mediating role of job involvement in this relationship. The findings are expected to provide both theoretical foundations and practical insights for policies and management practices aimed at stabilizing the caregiving workforce.

## 2. Theoretical Background and Hypotheses Development

### 2.1. Social Exchange Theory

Regarding the issue of caregiver retention intention, Social Exchange Theory (SET) can provide useful insight into the processes of interaction and the actions and outcomes between persons and organizations [20]. Developed from Thibaut and Kelley [21], the theory suggests that rational individuals will consider exchanging if they could evaluate the costs and benefits subjectively [22]. Cropanzano and Mitchell [23] further asserted that the establishment of interpersonal relationships is often grounded in a psychological mechanism that seeks to maximize benefits and minimize costs, and that imbalance in exchange relationships can lead to the erosion of trust and deterioration of relationships [24]. Such relationships are often founded on the principles of reciprocity and trust.

When exploring the effects of reciprocity and trust, SET plays an important role [25]. When organizations provide tangible or psychological support, employees feel valued, which in turn fosters a sense of obligation and loyalty toward the organization [26]. These positive exchange experiences help strengthen the foundation of mutual trust and promote long-term and stable interactions [27]. For example, the concept of perceived organizational support put forth by Eisenberger et al. [28] has empirical data in support of this mechanism, demonstrating that when employees perceive the organization cares about their benefits and values their contributions, employees are willing to respond with more responsibility and engagement to meet organizational standards. Nerstad et al. [29] have also noted that enabling organizational resources can be converted into employees’ organizational commitment and reciprocal actions, further fostering organizational effectiveness and strengthening the social exchange framework between both parties. Based on these reasons, this study considers SET to be a comprehensive explanatory framework for understanding the drivers of employees’ positive behaviors.

### 2.2. Organizational Commitment

Organizational commitment refers to employees’ emotional commitment, identification, and willingness to be invested in their organization and emphasizes the psychological connection and loyalty established during the interaction between people and organizations [30]. Meyer and Allen [31] broke organizational commitment down into three subdimensions: affective commitment, continuance commitment, and normative commitment. Affective commitment is the degree of emotional attachment stemming from recognition and a sense of belonging to the organization; continuance commitment refers to considering and evaluating the costs of leaving, ultimately leading individuals to remain with the organization after weighing the gains and losses; and normative commitment is characterized by feeling a sense of obligation stemming from moral responsibility and the internalization of values [32]. However, this study treats organizational commitment as a single construct because most prior literature on organizational commitment-related issues, particularly recent studies such as Azmy et al. [8], Ramaswamy et al. [32], and Redondo et al. [33], has also examined organizational commitment as a unidimensional construct. Furthermore, earlier studies on organizational commitment, such as Gutierrez et al. [34] and Tumwesigye [35], indicated that although organizational commitment can be divided into three sub-dimensions, the differences in their respective effects on behavior are not substantial. Since this study does not focus on the relative importance of the sub-dimensions of organizational commitment, organizational commitment is conceptualized and examined as a unidimensional construct.

Caregivers work in environments characterized by high emotional demands and intensive interpersonal interactions. When they perceive alignment between organizational culture and their personal values, it becomes easier for them to establish emotional attachment and identification [36]. At the same time, in institutions such as Veterans Homes that are highly mission-driven and ethically oriented, the role of normative commitment is particularly critical. Caregivers may choose to remain engaged in care work out of a sense of responsibility toward service recipients and moral identification with the organizational mission [37]. Furthermore, when caregivers perceive that leaving the organization would result in career disruption, loss of social support, or economic instability, continuance commitment becomes an important motivational factor influencing their retention behavior [38]. In this regard, organizational commitment not only reflects caregivers’ emotional tendencies and psychological intentions but also shapes their patterns of response to the overall value system of Veterans Homes.

Most prior studies have focused on nurses in medical or long-term care institutions [32] and general administrative staff [8]. Yet, for Veterans Homes, which have a specialized mission, differences in workforce demographics, task-driven culture, and institutional resources have yet to be properly examined. Caregivers face ongoing challenges such as negative work conditions, emotional fatigue, and instability in employment [3]. How stable commitment relationships can be established under conditions of limited organizational support or constrained professional development remains a critical issue requiring greater attention.

### 2.3. Work Value

Work value refers to the extent to which individuals attach importance to work itself and the rewards it brings, serving as the evaluative standard by which employees determine what is “right” or “important” [39]. Rokeach [40] defined it as individuals’ enduring beliefs about specific modes of conduct and regarded it as a core factor in constructing meaning from work experiences and forming behavioral tendencies [41]. Such beliefs are deeply embedded in individuals’ cognition of career goals, influencing how they choose and evaluate work roles, interpret workplace feedback, and decide whether to remain engaged in their work [42].

Given that caregivers in Veterans Homes must manage long-term interpersonal care relationships and heavy emotional labor [3], employees with strong intrinsic work values, such as autonomy, a sense of contribution, and the pursuit of self-actualization, are more likely to attain psychological fulfillment from caregiving practices, thereby enhancing organizational identification and work persistence [41]. Conversely, when employees emphasize extrinsic work values such as job stability, salary, and welfare benefits, the institutional guarantees provided by the organization become a crucial basis for determining whether to remain [13]. Notably, intrinsic and extrinsic values are not mutually exclusive orientations; individuals may place high importance on both, and this value combination shapes their overall evaluation of work and patterns of engagement [14].

Moreover, work value is influenced by the social structure and labor conditions in which caregivers are situated. For example, under circumstances characterized by long working hours, limited promotion opportunities, and emotional labor pressure, employees may regard security, stability, and protection as the primary sources of work meaning [43]. In contexts where caregivers commonly experience marginalization of social status and ambiguity of professional identity, the realization of extrinsic work values directly affects their retention decisions and psychological adjustment capacity [44]. In addition, autonomy-supportive leadership can strengthen intrinsic values, whereas controlling management styles may lead to greater reliance on extrinsic values [45].

Although work value is closely associated with several workplace variables such as job satisfaction, career choice, and job involvement [42,46], previous research has primarily focused on teachers [41] and technical professions [47]. Research on value structures in specific occupations, such as caregivers, remains relatively scarce. Therefore, the work value of caregivers represents a critical gap that this study seeks to address.

### 2.4. Retention Intention

Retention intention is defined as employees’ intention or inclination to stay with a job position for the long term, representing their degree of attachment to the organization and the level of future commitment they are willing to pursue [48]. This concept essentially reflects a positive work attitude, is inversely related to turnover intention, and is widely applied as an indicator for assessing workforce stability within organizations [49]. Particularly in care settings characterized by long-term demands, high emotional labor, and ethical pressures [43], whether caregivers are willing to remain in their positions becomes a critical factor influencing the quality of care.

Compared with general long-term care institutions, caregiving work in Veterans Homes involves not only physical care tasks but also the expectation of providing emotional support and social concern. In this situation, employees’ willingness to stay often depends not just on economic rewards and institutional guarantees, but also more internally on their work values, their sense of commitment to the organization, and on the emotional bonds they feel with recipients of care. Studies have shown that extrinsic conditions such as pay, job security, leadership support, and career development have an impact on employee retention tendencies [50,51]. Yet, without intrinsic motivations consistent with personal values, retention behavior will ultimately be unstable.

According to SET, individuals make rational choices in social exchanges based on subjective evaluations of costs and rewards [22]. Employees who perceive organizational investment in resources, support, or emotions respond with greater commitment [52]. Organizational commitment, as a prime reciprocal representation within social exchange, captures an employee’s intention by showcasing their desire to remain an active participant in the organization, desire to maintain strong relationships, and contribute to efforts [32]. Other studies suggested organizational commitment is a good predictor of retention and turnover intentions [7,10]. As such, this study proposes the following hypothesis:

**H1:** 
*Organizational commitment has a positive effect on retention intention.*


When employees seek not only external rewards but also the realization of their core personal values and the meaning of their careers within an organization, stronger work values can be developed [41]. When the organization fulfills employees’ expectations for autonomy, a sense of achievement, or social contribution, a situation of value congruence and exchange balance is formed, thereby encouraging them to demonstrate a tendency to remain [13]. In workplaces where caregivers are exposed to long-term stress and emotional exhaustion, the extent to which intrinsic and extrinsic values are realized directly affects their retention intention [53]. In recent years, numerous studies have supported this argument [54,55]. Therefore, this study proposes the following hypothesis:

**H2:** 
*Work value has a positive effect on retention intention.*


### 2.5. Job Involvement

Job involvement is widely defined as the extent to which employees identify with, concentrate on, and psychologically invest in their work, and it is considered part of one’s self-concept [56,57]. Job involvement not only represents the time and energy employees devote to their duties but also reflects an extension of their values and self-image. When work is regarded as a central aspect of life, employees tend to demonstrate greater focus and persistence [58,59]. In Veterans Homes, caregivers face highly intensive work and significant emotional labor. If they view caregiving as part of their self-value, they are more likely to continue showing responsibility and patience in the face of stress and challenges, thereby maintaining loyalty to the organization.

In previous studies, job involvement has been widely examined in various contexts such as education, healthcare, and government sectors. However, most of this literature has focused on service quality, organizational effectiveness, or job performance as key outcome variables, with relatively few studies considering “retention intention” as a core outcome [59,60]. In the context of long-term care, whether caregivers are willing to remain in their positions often reflects organizational stability and service quality more accurately than short-term performance. Nonetheless, existing research has paid insufficient attention to the central issue of “whether employees choose to stay,” leaving a clear gap in both theoretical and practical discussions.

SET emphasizes that interpersonal and organizational relationships are built on reciprocity and a cost–benefit evaluation [22]. When employees perceive that the organization invests resources in support and care, such as providing stable working conditions, respecting professional values, and fostering a sense of belonging, they are inclined to respond with greater involvement [61]. For caregivers in Veterans Homes, the emotional attachment and loyalty fostered by organizational commitment encourage them to regard themselves as integral members of the organization and to devote greater effort to daily caregiving. This relationship between commitment and involvement aligns with the perspective of SET: the organization provides support and value recognition, while employees reciprocate with high involvement, thereby reinforcing the exchange relationship. Based on this reasoning, the following hypothesis is proposed:

**H3:** *Organizational commitment has a positive effect on job involvement*.

With regard to work value, from the perspective of SET, interactions between employees and the organization are based on reciprocity [26]. When caregivers perceive that the organization respects and responds to their core work values through job design or resource support, such as providing a safe workplace, fair compensation, or opportunities for self-development, these inputs are interpreted as organizational goodwill and commitment. According to SET, employees tend to maintain exchange balance through reciprocal behaviors, which manifest as higher levels of job involvement [20].

Given that caregiving work in Veterans Homes is highly labor-intensive and emotionally demanding, if caregivers’ intrinsic or extrinsic work values are fulfilled, they are more likely to generate positive reciprocity through job involvement. This finding is consistent with prior research suggesting that employees demonstrate greater involvement and loyalty when they perceive that their needs and values are met by the organization [60,62]. Therefore, the following hypothesis is proposed:

**H4:** *Work value has a positive effect on job involvement*.

According to SET, the interaction between individuals and organizations is built on trust, reciprocity, and a sense of obligation [20]. When caregivers in Veterans Homes perceive that the organization values their contributions—for example, through fair work arrangements, emotional support, and opportunities for professional growth—they interpret these actions as part of the social exchange and respond with greater job involvement. This involvement serves as a form of positive psychological capital, reinforcing caregivers’ sense of belonging to the organization and their intention to continue serving in the long term [63].

Previous research has pointed to a strong positive relationship between job involvement and retention intention. Employees with high job involvement tend to show stronger organizational loyalty, lower turnover intention, and greater resilience and efficiency under stress [64,65]. Moreover, when caregivers regard caregiving as a means of fulfilling intrinsic values and achieving self-identity, they are more likely to develop solid organizational commitment, which in turn enhances their retention intention. This aligns with the “principle of reciprocity” in SET, which posits that in high-quality social exchange relationships, individuals choose to remain and continue contributing in response to organizational support [52].

Job involvement may also enhance retention intention by improving job satisfaction, reducing burnout, and fostering emotional bonds, all of which strengthen employees’ positive attitudes toward work [64]. When work is no longer merely a means of livelihood but becomes a domain of self-realization and social contribution, employees are inclined to remain in the field for long-term development [66]. Particularly in caregiving tasks with deep emotional significance, such as caring for veterans, caregivers with strong job involvement are more likely to generate a sense of mission and ethical responsibility, which motivates them to stay.

Based on the above theoretical and empirical evidence, SET provides a robust framework for explaining how caregivers shape their organizational identification and long-term retention intention through job involvement. Therefore, the following hypothesis is proposed:

**H5:** *Job involvement has a positive effect on retention intention*.

Based on the above logical reasoning, if organizational commitment and work value can effectively enhance caregivers’ job involvement, this mediating process may further transform into long-term commitment and retention behavior toward the organization. In other words, job involvement is not only an expression of attitude but may also serve as a critical mechanism linking antecedents and outcomes. Therefore, this study proposes the following hypotheses:

**H6:** *Organizational commitment positively influences retention intention through job involvement*.

**H7:** *Work value positively influences retention intention through job involvement*.

Drawing upon the literature review above, this study formulated the research model depicted in Figure 1.

## 3. Research Methodology

### 3.1. Sample and Procedure

This cross-sectional study adopted a quantitative design. Since the author is employed at a Veterans Home, contact was made with 16 Veterans Homes by telephone to explain the research purpose, request assistance in recruiting 50 frontline caregivers as participants, and arrange a questionnaire distribution schedule. Data were collected through a paper-based on-site questionnaire between September and November 2024. After completing the survey, caregivers returned the questionnaire to the researcher, who immediately sealed it in an envelope to ensure privacy and anonymity. During the data entry process, questionnaires with incomplete responses were excluded to secure the integrity and validity of the dataset. Based on the number of measurement items and the rules for sample size estimation, the study aimed to collect at least 450 valid responses.

Since some caregivers expressed reservations about completing the questionnaire, 540 questionnaires were collected. After eliminating responses with missing values and invalid answers, 477 valid questionnaires remained. Regarding demographic characteristics (see Table 1), most respondents were female, with 366 individuals accounting for 76.7 percent. In terms of age, most respondents were between 51 and 60 years old (151 individuals, 31.7 percent), followed by those aged 61 to 65 years (92 individuals, 19.3 percent). With respect to seniority, the largest group had between 5 and 15 years of experience (169 individuals, 35.4 percent), followed by those with 2 to 5 years (124 individuals, 26.0 percent). Regarding education, most respondents had a high school or vocational school education or above; for example, 229 had a high school or vocational school or higher (48.0 percent). The majority of respondents were married, 255 respondents in total (53.5 percent). In terms of economic responsibility, 61.8 percent of the respondents indicated they were the primary income provider for the household. Household income was most concentrated in the range of TWD 28,001 to TWD 35,000, (TWD = New Taiwan Dollar), with 232 respondents (48.6 percent).

### 3.2. Instrument

The questionnaire design of this study was divided into five main sections, measuring organizational commitment, work value, job involvement, retention intention, and respondents’ demographic information. To ensure the reliability and validity of the questionnaire, all measurement items for each construct were adapted from the relevant empirical literature and adjusted appropriately to fit the research context. All items were measured using a five-point Likert scale, where 1 represented “strongly disagree” and 5 represented “strongly agree,” in order to quantify respondents’ level of agreement.

The first section, “Organizational Commitment,” consisted of nine items adapted from the study of Murray and Holmes [67], with the questionnaire revised from Meyer et al. [68]. The second section, “Work Value,” included 17 items modified from the scale developed by Lin et al. [69], which was originally developed based on Hackman and Oldham [70]. The third section, “Job Involvement,” contained 10 items drawn from the scale proposed by Kanungo [57], which was the original instrument. The fourth section, “Retention Intention,” comprised seven items adapted from the research of Chen and Chen [71], with the questionnaire revised from Tsai and Chang [72]. Among the items, OC5 and JI2 were reverse-coded, and the data were processed accordingly. Following the back-translation method [73], a preliminary Chinese version of the questionnaire was developed and reviewed by two academic experts and one field expert to ensure content validity and to make necessary revisions. The final section addressed personal demographic variables, including gender, age, and working experience. These three demographic variables were also included as moderating variables, as previous studies have shown that such individual factors influence retention intention [74,75]. The complete scale can be found in Appendix A.

### 3.3. Ethical Considerations

This research was approved and cleared by the Quantitative Analysis and Research Association (No. 1130705001).

### 3.4. Data Analysis

The first author analyzed the collected data using structural equation modeling (SEM), with Amos statistical software version 24 employed as the analytical tool to examine the variables. The data processing first involved testing for univariate normal distribution, following Kline’s [76] recommendation that the absolute value of skewness should be less than 2 and the absolute value of kurtosis should be less than 7. Path analysis was then conducted. Finally, for the mediator analysis, the bootstrapping method proposed by Hayes [77] was applied, with 5000 resamples used to assess the indirect relationships among the structural paths.

### 3.5. Common Method Bias

As this study collected data through a self-reported questionnaire survey, the potential for common method bias (CMB) requires attention [78]. For procedural remedies, following the recommendations of Podsakoff et al. [79], anonymity of respondents was ensured, and all scale items were pretested and adapted from established studies to reduce the influence of CMB. In addition, items from different constructs were randomly mixed in the questionnaire, and the research purpose and variable names were concealed to enhance the authenticity of responses [80].

For post hoc testing, Harman’s single-factor test was first conducted to assess the extent of CMB. The unrotated factor analysis results showed that the first principal component accounted for only 38.4% of the variance, which did not exceed the 50% threshold. Second, following the approach recommended by Podsakoff et al. [79], a common latent factor was included, and the explained variance of substantive constructs and the method factor for each item was examined. The results indicated that the average substantive explained variance was 0.549, while the average method variance was only 0.020, with a ratio of approximately 27:1. Overall, these results suggest that the impact of CMB in this study was not severe.

## 4. Results

The analysis followed the two-step approach of Anderson and Gerbing [81] to test the measurement and structural models. In the first step, confirmatory factor analysis and discriminant validity assessment were conducted to examine the reliability and validity of the sample. In the second step, structural model analysis was performed. After assessing model fit, the structural paths were analyzed to test the relationships among the constructs.

### 4.1. Measurement Model

Before conducting the measurement model analysis, the sample data were first tested for univariate normal distribution. The statistical results indicated that the absolute values of skewness for all variables ranged from 0.031 to 1.651, and the absolute values of kurtosis ranged from 0.000 to 2.955. All values ranged within the recommended guidelines of Kline [76], indicating that the data met the condition of univariate normality and were appropriate for the following SEM analysis. A confirmatory factor analysis was conducted to assess the measurement model’s convergent validity. According to Hair et al.’s [82] guidelines, as long as the standardized factor loadings of each of the items are greater than 0.6, the composite reliability (CR) of the latent construct is greater than 0.7, and the average variance extracted (AVE) greater than 0.5, the constructs can be assumed to have good convergent validity.

In this study, item 5 of the “Work Value” construct (WV5) had a factor loading of 0.495, which was below the acceptable threshold, and was therefore removed. After removing WV5, the α, CR, and AVE values were recalculated. Overall, as shown in Table 2, the results after deleting WV5 indicated that all constructs met the evaluation standards recommended by Hair et al. [82], demonstrating that the latent variables in this study exhibited good measurement consistency and convergent validity.

Second, Hair et al. [82] suggested that if the absolute value of the Pearson correlation coefficient between constructs is smaller than the square root of the AVE of each construct, discriminant validity can be considered adequate. As shown in Table 3, the absolute values of the Pearson correlation coefficients among all constructs (presented in the lower triangle) were smaller than the square root of the AVE values on the diagonal, indicating that the measurement scales in this study exhibited good discriminant validity. In addition, Kline [76] noted that if the absolute value of the Pearson correlation coefficient between constructs is less than 0.850, the constructs can be regarded as distinguishable. Based on the data of this study, the correlation coefficients among the constructs ranged from 0.333 to 0.674, which were clearly below the threshold, further supporting the conclusion that the constructs demonstrated good discriminant validity. In addition to using the Fornell–Larcker criterion, this study also applied the HTMT standard proposed by Henseler et al. [83]. According to their recommendation, the HTMT value should be lower than 0.85. As shown in the upper-right triangle of Table 3, the HTMT values in this study ranged from 0.370 to 0.739, all below the threshold of 0.85. Therefore, discriminant validity was further confirmed through HTMT.

### 4.2. Structural Model

Based on the model fit analysis results, the Bollen–Stine χ^2^ for the sample was 1125.226 with 813 degrees of freedom; the χ^2^/DF was 1.384, within an accepted range for this type of analysis. The other fit indices indicated that the results were also satisfactory: RMSEA = 0.028; SRMR = 0.067; TLI = 0.977; CFI = 0.978; IFI = 0.978; Hoelter’s N = 344.982; and all of the fit indices met or were close to ideal levels suggested by Bagozzi and Yi [84] and West et al. [85]. Overall, all the fit indices suggested that the structural models of the current study demonstrated a good fit and were adequate representations of the structural relationships of the population of interest.

The results of the structural model analysis are presented in Table 4 and Figure 2. The effect of organizational commitment on retention intention was positive and significant (β = 0.254, *p* < 0.001), supporting H1. The effect of work value on retention intention was also significantly positive (β = 0.169, *p* < 0.001), providing further support for H2. In addition, the effect of organizational commitment on job involvement was highly significant (β = 0.619, *p* < 0.001), supporting H3, while the positive effect of work value on job involvement was also significant (β = 0.347, *p* < 0.001), supporting H4. Furthermore, the effect of job involvement on retention intention was positive and significant (β = 0.401, *p* < 0.001), supporting H5.

Regarding mediating effects, organizational commitment indirectly influenced retention intention through job involvement, and this effect was significant (β = 0.248, *p* = 0.001), supporting H6. Similarly, work value also had a significant positive indirect effect on retention intention through job involvement (β = 0.139, *p* < 0.001), supporting H7. Overall, all hypotheses were statistically supported, and the significance of both direct and indirect effects was further confirmed through the bootstrapping procedure with 5000 resamples. These results demonstrate that the research model possesses strong explanatory power and robustness.

## 5. Discussion

### 5.1. The Effects of Organizational Commitment on Job Involvement and Retention Intention

The results of this study indicate that organizational commitment has a significant positive effect on both job involvement and retention intention. Among these, the effect of organizational commitment on job involvement is one of the strongest, suggesting that when employees develop identification with and loyalty to the organization, they are more likely to demonstrate vigor, dedication, and absorption in their work. Furthermore, organizational commitment directly enhances retention intention, showing that when caregivers possess a stronger emotional attachment to or a sense of responsibility toward Veterans Homes, they are more willing to continue contributing to the organization and maintain their job positions.

According to SET, the relationship between employees and organizations is an exchange process. When employees are given support, respect, and resources from the organization, they return this by being loyal, involved, and providing service to the organization [20]. Organizational commitment is the tangible expression of this exchange relationship. Its meaning extends beyond recognition of organizational goals, reflecting instead a psychological tendency for employees to actively respond to organizational expectations [37]. For Veterans Home caregivers, when tangible recognition and psychological support are provided in the context of high emotional labor and stress, their emotional attachment to the institution is strengthened, which translates into greater engagement in daily work and stronger retention intention.

These results also align with the views of Marcoux et al. [37], who claimed that employees stay with organizations because they “want to.” That type of commitment, based on identification, values, and emotions, is more effective in promoting job involvement than commitment developed from moral obligation or costs of leaving. Particularly in caregiving contexts where clear promotion opportunities and material incentives are limited, organizational commitment and intrinsic work motivation serve as the core sources for maintaining workforce stability.

### 5.2. The Potential Meaning and Influence of Work Value Should Be Emphasized

The findings from this study indicate that work value has a significant positive influence on caregivers’ job involvement and retention intention. Specifically, accomplishing caregiver’s work values—autonomous functioning, sense of achievement, and social contribution—is important for increasing job involvement, affecting retention likelihood. This is in keeping with the notion described by Busque-Carrier et al. [44]. From the perspective of SET, when individuals perceive that their core values are practiced and affirmed in their work, they regard this as a psychological resource or feedback provided by the organization, which fosters reciprocal behavioral intentions [22]. This exchange relationship, built upon respect for personal values and meaning fulfillment, leads caregivers to reciprocate with greater engagement and loyalty, forming stable affective commitment and stronger retention intention [52]. This type of exchange facilitates job involvement and influences caregivers’ commitment and motivation to stay.

In practice, work value is often distilled into externally valued rewards like salary and benefits. Ede et al. [41] note that employees still have a deep yearning for “meaningful work.” Kalleberg and Marsden [39] also argued that social institutions and economic contexts influence work values. Particularly in high-pressure and high-demand caregiving environments such as Veterans Homes, caregivers may value stability and compensation. However, reliance on extrinsic incentives alone cannot sustain their work enthusiasm and organizational commitment. According to SET, if organizations provide only material exchanges, employees are prone to developing transactional attitudes toward work, lacking emotional identification and long-term engagement [24]. Conversely, when work values are recognized and fulfilled by the organization, caregivers’ intrinsic identification and sense of achievement are satisfied, and their job involvement and retention intention are also strengthened [13]. Therefore, from the perspective of work value, Veterans Homes can reevaluate the design of caregiver motivation and support systems, which may help address turnover challenges and reinforce positive interaction between the organization and its employees.

### 5.3. The Core Mediating Role of Work Engagement in Retention Intention

The results from this study indicate that job involvement is a key mediating mechanism. SET argues that individuals engage positively in return for an organization when they believe it provides them with resources, support, goodwill, and investment [86]. In this context, when caregivers feel respected and trusted by Veterans Homes or their work values are fulfilled, they are likely to experience psychological involvement and participation. This, in turn, is expressed through job involvement as a positive response to the organization, which subsequently fosters retention intention.

Another often overlooked aspect is that caregivers work under conditions characterized by long working hours, intensive emotional labor, and high ethical stress. Their retention intention is not solely determined by material rewards but is more strongly driven by emotional identification and intrinsic value perceptions [87]. When Veterans Homes provide caregivers with respect for their professionalism, create a supportive environment, strengthen their organizational commitment, or enable the realization of their work values, caregivers are more likely to invest emotionally in their roles, generating strong motivation and focused behavior [22]. Such engagement further increases retention intention [88].

Therefore, if Veterans Homes hope to deal with the challenge of workforce shortages effectively, based on the findings of this study, raising salaries or offering basic benefits to caregivers is not the only way. At the same time, it is imperative to understand on a deeper level what caregivers engage with and are attached to, as caregivers in high-stakes caregiving situations, and to clearly understand the values caregivers pursue within their roles and in high-stress situations. Creating an environment that promotes the engagement of caregivers will result in stronger retention intention.

### 5.4. Theoretical Implications

First, based on SET, this study explains how caregivers’ retention intention in Veterans Homes is formed through psychological exchange processes. Previous literature has largely focused on nurses in medical institutions or general long-term care facilities, with less attention paid to the unique labor context shaped by the special system and service recipients of Veterans Homes [18,19]. This study extends the application of SET by verifying how organizational commitment and work value influence retention intention through job involvement, thereby addressing the theoretical gap regarding how exchange relationships transform into stable retention behaviors.

Second, this study reveals that organizational commitment has a significant impact on job involvement and retention intention, indicating that exchange motives driven by affective attachment and organizational identification can effectively enhance caregivers’ willingness to continue providing services. This finding responds to the limitations of existing studies that mainly emphasize extrinsic incentives, such as salary and benefits [60,62], and underscores the importance of intrinsic emotional bonds in long-term care contexts. It thus provides a more comprehensive theoretical perspective for understanding caregivers’ retention behaviors.

Finally, this study demonstrates that job involvement serves as a key mediating role between organizational commitment, work value, and retention intention. This extends the theoretical assumptions of reciprocity and return within SET [22], elevating job involvement from a mere positive psychological state to a mediating process that embodies the exchange mechanism. In other words, when caregivers perceive organizational support and the realization of their values, they respond with higher levels of job involvement, which in turn strengthens their intention to remain. This result not only enriches the understanding of retention intention mechanisms in the field of organizational behavior but also provides a theoretical foundation for future research.

### 5.5. Practical Implications

First, managers can concentrate on developing positive social exchange relationships in order to encourage caregivers’ job involvement and stability. Over time, employees will feel positive commitment toward the organization when managers are fair to employees, provide emotional support, acknowledge work, and continually show gratitude. In particular, for caregivers working long-term in high-pressure and emotionally demanding environments, organizations should approach from the perspective of being “seen” and “understood,” thereby enhancing their enthusiasm for work and willingness to continue serving, which in turn promotes workforce stability and improves the overall quality of care.

Furthermore, job involvement represents a strategically significant management indicator for Veterans Home caregivers, who often face staff shortages and heavy emotional burdens [89]. This study suggests managers should emphasize organizational commitment and mechanisms that align with caregivers’ values. Caregivers’ engagement can be stimulated through clear career development pathways, emotional support systems, and the design of meaningful work tasks. In particular, when supervisors demonstrate care and provide feedback, it helps cultivate high-quality social exchange relationships, reinforcing stable retention intention [90].

### 5.6. Limitations and Future Research

A primary benefit of this study was the sample. In contrast to prior literature that has predominantly examined hospitals or general long-term care facilities, this study took a unique approach by examining caregivers in Veterans Homes—a distinct type of facility with its own institutional mission and client recipient group. This study had 477 valid responses from all 16 Veterans Homes across Taiwan, which offered a broad dataset and representativeness. This study contributes to the literature by addressing a gap and presenting practical insights that are timely and valuable for workforce stability and sustainability in the veterans’ long-term care system.

Several limitations were evident in this study. First, the cross-sectional design of the study limits attribution of causation. Second, using self-reported questionnaires to potentially create common method bias from the on-site data collection with the authors present. For future research and to alleviate this concern, the questionnaires could be placed into sealed drop boxes after completion, within a given time frame, and researchers could collect the boxes afterwards for different levels of anonymity. Third, the non-probability sampling design impacts the generalizability of the results, and clustering of values in Veterans Homes was not taken into account, potentially producing inaccuracies in parameter estimates. Fourth, there could be construct specification issues, as different approaches to operationalizing organizational commitment, work value, or retention intention could each produce unique effects on associations. Lastly, this study focused on one industry as well as an industry with a limited demographic focus. Using longitudinal designs, probability sampling, or multi-level analysis could add rich content to the existing research, in addition to examining a wider-scope sample covering other occupational groups and cultural contexts. Also, implementing work motivation for further explanatory purposes would assist further study in establishing a robust and comprehensive theoretical model.

## 6. Conclusions

This study applied SET to explain caregivers’ psychological exchange behaviors and to examine how organizational commitment and work value influence their retention intention through job involvement. The findings reveal an important insight: not only does organizational commitment directly and indirectly affect retention intention, but work value also indirectly promotes caregivers’ retention intention by enhancing job involvement. This discovery highlights that, in addition to strengthening institutional support and emotional connections, managers should pay greater attention to employees’ value recognition of work and their level of engagement. When caregiving institutions effectively guide employees to develop positive emotional involvement, retention intention can be significantly enhanced. Furthermore, the results reaffirm that SET provides a robust framework for explaining caregivers’ psychological processes, offering valuable insights for improving workforce stability in long-term care institutions.

## Figures and Tables

**Figure 1 healthcare-13-02396-f001:**
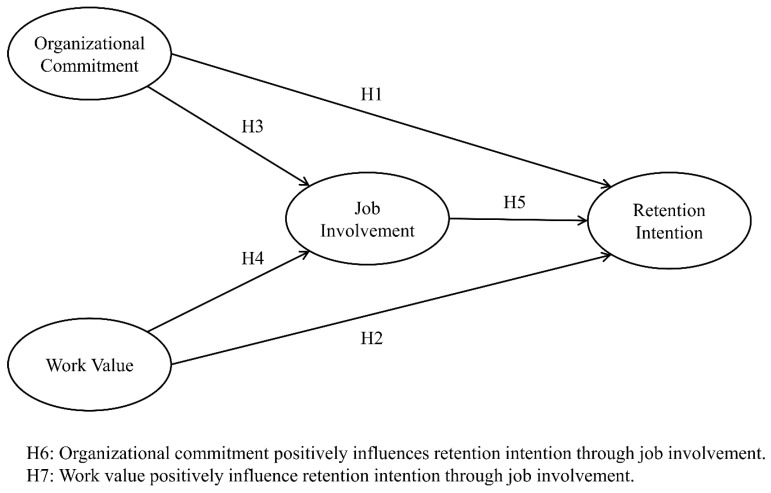
Conceptual model.

**Figure 2 healthcare-13-02396-f002:**
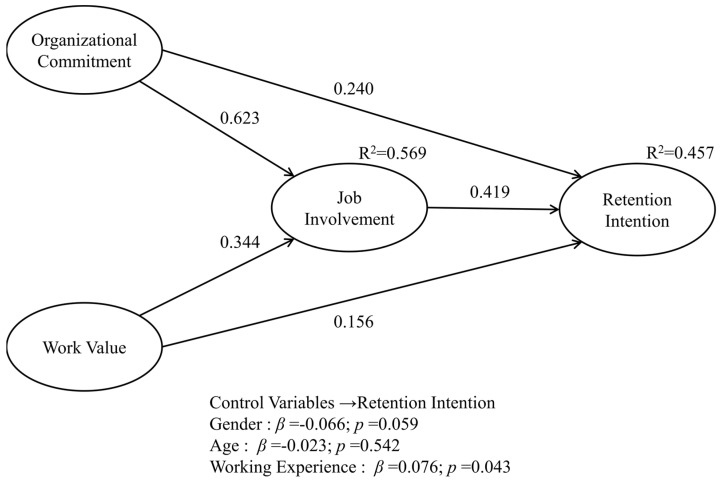
Results of theoretical framework. Note: the values represent standardized regression coefficients.

**Table 1 healthcare-13-02396-t001:** Distribution of demographic characteristics.

Demographic Characteristics (*N* = 477)	Category	Frequency (*n* = Responses)	Percentage (%)
Gender	Male	111	23.3
	Female	366	76.7
Age	21–30 years	32	6.7
	31–40 years	30	6.3
	41–50 years	97	20.3
	51–60 years	151	31.7
	Above 61 years	167	35.0
Seniority	Within 6 months	35	7.3
	6 months to 1 year	21	4.4
	1 year to 2 years	64	13.4
	2 years to 5 years	124	26.0
	5 years to 15 years	169	35.4
	15 years to 20 years	44	9.2
	More than 20 years	20	4.2
Education level	Junior high school or below	133	27.9
	Senior/vocational high school	229	48.0
	College	62	13.0
	Graduate	53	11.1
Marital status	Married	255	53.5
	Divorced	79	16.6
	Widowed	56	11.7
	Unmarried	87	18.2
Economic provider	Yes	295	61.8
	No	182	38.2
Household income	Below 28,000 NTD	37	7.8
	28,001 to 35,000 NTD	232	48.6
	35,001 to 42,000 NTD	113	23.7
	42,001 to 50,000 NTD	36	7.5
	Above 50,001 NTD	59	12.4
*n* = 477			

**Table 2 healthcare-13-02396-t002:** Confirmatory factor analysis and scale reliability.

Items	Unstd.	S.E.	t	*p*	Std.	α	CR	AVE
Organizational Commitment (OC)						0.904	0.907	0.521
OC1	1				0.745			
OC2	0.732	0.053	13.750	<0.001	0.635			
OC3	0.785	0.052	14.960	<0.001	0.686			
OC4	0.714	0.051	13.984	<0.001	0.645			
OC5	0.934	0.060	15.609	<0.001	0.714			
OC6	1.314	0.075	17.446	<0.001	0.790			
OC7	1.100	0.066	16.786	<0.001	0.763			
OC8	1.144	0.075	15.284	<0.001	0.700			
OC9	1.099	0.063	17.590	<0.001	0.796			
Work Value (WV)						0.950	0.951	0.552
WV1	1				0.673			
WV2	1.088	0.082	13.194	<0.001	0.647			
WV3	1.084	0.076	14.318	<0.001	0.708			
WV4	1.116	0.079	14.153	<0.001	0.699			
WV6	1.153	0.079	14.600	<0.001	0.723			
WV7	1.144	0.077	14.854	<0.001	0.737			
WV8	1.195	0.076	15.810	<0.001	0.791			
WV9	1.235	0.078	15.821	<0.001	0.792			
WV10	1.168	0.073	15.927	<0.001	0.798			
WV11	1.395	0.091	15.327	<0.001	0.764			
WV12	1.259	0.083	15.134	<0.001	0.753			
WV13	1.275	0.078	16.446	<0.001	0.828			
WV14	1.279	0.081	15.756	<0.001	0.788			
WV15	1.184	0.077	15.467	<0.001	0.772			
WV16	1.328	0.093	14.285	<0.001	0.706			
WV17	1.214	0.088	13.789	<0.001	0.679			
Job Involvement (JI)						0.911	0.913	0.513
JI1	1				0.706			
JI2	1.164	0.083	14.018	<0.001	0.673			
JI3	1.596	0.102	15.633	<0.001	0.752			
JI4	1.443	0.095	15.191	<0.001	0.730			
JI5	1.215	0.086	14.145	<0.001	0.679			
JI6	1.604	0.100	16.015	<0.001	0.771			
JI7	1.752	0.110	15.935	<0.001	0.767			
JI8	1.280	0.090	14.269	<0.001	0.685			
JI9	1.124	0.076	14.813	<0.001	0.712			
JI10	1.107	0.078	14.159	<0.001	0.680			
Retention Intention (RI)						0.926	0.928	0.650
RI1	1				0.801			
RI2	1.003	0.050	20.194	<0.001	0.814			
RI3	1.131	0.062	18.369	<0.001	0.758			
RI4	1.093	0.059	18.524	<0.001	0.763			
RI5	0.980	0.047	20.752	<0.001	0.830			
RI6	0.980	0.047	21.053	<0.001	0.839			
RI7	1.150	0.055	20.858	<0.001	0.833			

**Table 3 healthcare-13-02396-t003:** Discriminant validity assessment.

Constructs	Mean	SD	AVE	Discriminant Validity			
				OC	WV	JI	RI
Organizational Commitment (OC)	3.598	0.478	0.521	**0.722**	0.370	0.739	0.611
Work Value (WV)	4.338	0.497	0.552	0.333	**0.743**	0.585	0.484
Job Involvement (JI)	3.864	0.605	0.513	0.674	0.531	**0.716**	0.683
Retention Intention (RI)	4.051	0.700	0.650	0.566	0.450	0.631	**0.806**

Note: (1) The diagonal bold and gray background numbers represent the square root of AVE; (2) the lower-left triangle shows the correlation coefficients; (3) the upper-right triangle shows the HTMT.

**Table 4 healthcare-13-02396-t004:** Path coefficients and significances.

Path Analysis	Std.	Unstd.	S.E.	*p*	Supported
Control Variables					
Gender → Retention Intention	−0.066	−0.101	0.054	0.059	N
Age → Retention Intention	−0.023	−0.011	0.018	0.542	N
Working Experience → Retention Intention	0.076	0.034	0.017	0.043	Y
Hypothesis					
H1: Organizational Commitment → Retention Intention	0.240	0.332	0.087	<0.001	Y
H2: Work Value → Retention Intention	0.156	0.248	0.074	<0.001	Y
H3: Organizational Commitment → Job Involvement	0.623	0.583	0.049	<0.001	Y
H4: Work Value → Job Involvement	0.344	0.370	0.045	<0.001	Y
H5: Job Involvement → Retention Intention	0.419	0.619	0.109	<0.001	Y
H6: Organizational Commitment → Job Involvement → Retention Intention	0.261	0.361	0.087	0.001	Y
H7: Work Value → Job Involvement → Retention Intention	0.144	0.229	0.055	<0.001	Y

Note: Bootstrapping = 5000; Job Involvement R^2^ = 0.569 (0.517, 0.624); Retention Intention R^2^ = 0.457 (0.376, 0.552).

## Data Availability

The original contributions presented in this study are included in the article. Further inquiries can be directed to the corresponding author.

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
