# Peer review of "Why Organizational Commitment and Work Values of Veterans Home Caregivers Affect Retention Intentions: A Social Exchange Theory Perspective"

_healthcare, 2025, doi:10.3390/healthcare13192396_

Round 1
Reviewer 1 Report
Comments and Suggestions for Authors
The article addresses an issue in a specific context that, in my opinion, is in line with the humanization and personalization of healthcare. The study uses Social Exchange Theory as its theoretical basis, which is an interesting aspect. I congratulate the authors!
I have a few comments on how to improve the text presented.
Abstract:
• Present the objective of the study, as it is not mentioned.
Keywords:
• Their designation should be reviewed and reformulated taking into account the MESH designation.
Introduction:
• The works and authors in the literature that justify the statement made should be mentioned in order to enrich the bibliographic references presented. The objective and research question formulated for the study should also be presented more clearly.
Methodology:
• The first paragraph is insufficient and quite incomplete in contextualizing the paradigm and type of study. It should be developed using reference authors;
• The inclusion and exclusion criteria for participants in the intentional sample should be presented clearly and objectively;
• The form, stages, and person responsible for the intentional selection of participants and how the authors gained access to the participants;
• In point 3.2 (line 306) referring to the instrument, its sections should be developed;
• A point should be introduced for ethical issues related to the entire research process and, above all, related to access to participants and issues of their participation and informed consent. The information reported that consent was given verbally raises many concerns for me at a fundamental point in the research process.
Data analysis:
• This point should be developed. Who performed the data analysis?
Results:
• The note placed on the tables should be reworded;
Discussion and conclusions:
• Following on from what is presented in the results of the article, it seems important to highlight the individual aspects, which are mentioned but are important in terms of implications and results. I believe there should be a paragraph linking the theory used and the results presented with the practical implications of their results in light of the evidence presented.
Reviewer 2 Report
Comments and Suggestions for Authors
I’m not sure I saw a definition for ‘caregivers.’ Please provide a description as to job duties or a definition of a caregiver in a work facility such as the veteran homes in Taiwan.
If your reading audience is interested in this topic, your study has merit in the local setting. It is significant and needs to be presented in various venues. Kudos.
Line 290: ‘Since some caregivers expressed reservations about completing the questionnaire, 540 questionnaires were collected.’ How did you annotate the individuals who chose not to participate? Out of 800 potential participants who met the inclusion criteria (need to include criteria somewhere in methods), questionnaires were distributed, and Xxx completed the questionnaire successfully; however, others were eliminated based on exclusion criteria such as x.x.x. You have the data; it just needs to be included.
Line 302-303: include a definition for NT$28,001 to NT$35,000, (NT$ = New Taiwan Dollar)
In table 1, for reference, include N – 477 so we can see if your math is correct. A separate column for percentages would represent values.
|
Demographic Characteristics (N = 477) |
Category |
Frequency (n = responses) |
Percentage (%) |
Section 3.2 Instrument: where did the questionnaire come from before you made changes? Did you obtain 3rd party approval to use their instruments? Did you alter any of the questions? SMEs? Perform a field test for reliability & validity?
Line 321: The data were analyzed using structural equation modeling (SEM) with Amos statistical software.... did you have a specific version for this model? Just need to add here.
Line 496: Would a recommendation to conduct further studies in other countries be a consideration? What about any strengths of your study?
References: I recommend finding some current articles to support your discussion.
Reviewer 3 Report
Comments and Suggestions for Authors
Dear Authors,
Please, see below few comments you might consider or clarify:
-
Ethics approval & consent are inconsistent / inadequate
The paper states participants gave verbal consent during on-site administration, yet the “Informed Consent Statement” reads “Not applicable.” These cannot both be true for human-subjects research. Provide IRB/ethics approval (name, number, date) and correct the consent statement—or explain a formal waiver. -
Construct mis-specification of “work engagement.”
WE is defined using the UWES framework (vigor, dedication, absorption), but the instrument used is Kanungo (1982) job/work involvement, not UWES. This is a fundamental construct mismatch, risking invalid inferences (the items capture involvement/identification, not vigor/dedication/absorption). Either (a) replace WE with UWES; or (b) reframe the construct as job involvement throughout and reinterpret results. -
Questionable content validity: Work value items overlap with engagement/effort.
WV includes items such as “I am proud of my work,” “I am dedicated,” “I strive for perfection,” “I often arrive early,” “I’m willing to work overtime,” which align more with engagement or conscientiousness than with “values.” This overlap threatens discriminant validity and likely inflates paths WV→WE and WV→RI. Consider re-specifying WV (retain value-content items; drop engagement-like items) and re-running CFA/SEM; also report HTMT in addition to Fornell–Larcker -
Single-source, same-time data with no common-method bias (CMB) diagnostics.
All constructs are self-reported in one sitting; no procedural or statistical CMB remedies are described (e.g., marker variable, CFA single-factor, ULMC, or VIF for CMB). Add and report CMB checks. -
Clustered sampling ignored in analysis.
You sampled ~50 caregivers in each of 16 homes via purposive sampling but analyzed with single-level SEM, assuming independent observations. Intra-home clustering likely deflates SEs. Please report ICCs and re-estimate with cluster-robust SEs or multilevel SEM; at minimum, adjust SEs for clustering. -
Over-interpretation of path magnitudes without tests of difference & causal language.
The abstract and discussion assert OC’s effect on WE is “stronger” and has “greater implications,” but no formal tests of coefficient differences are reported. Also, cross-sectional data do not warrant causal claims for mediation. Rephrase to associational language and add Wald/bootstrapped contrasts for path differences if you wish to compare magnitudes. -
Factor loading threshold error.
You write that “standardized factor loadings… greater than 6” satisfy Hair et al. This is a typo; it should be > 0.6. Correct the text -
Item deletion not reflected in Appendix; incomplete item lists.
WV5 was removed due to low loading, but Appendix A.1 still lists WV5. Also, WE has 10 items in the method, yet the appendix shows only WE1–WE4; RI items are not listed at all. Update Appendix A.1 to match the final scales (each item, final wording, reverse-coding) -
Table 2 formatting/consistency errors.
In Table 2, multiple rows have misaligned/missing entries (e.g., WE1 shows only “1” and “< .001 .706” without Unstd/S.E./t values). Please correct all table columns and ensure they total consistently after item deletion. -
Reverse-scored items not documented.
Several items appear negatively worded (e.g., WE2 “only a small part of who I am”; OC5 “not yet made substantial contributions”; WV15 “never feel confused or afraid”). State which items were reverse-coded and confirm coding in CFA/SEM. -
No explained variance (R²) reported.
You claim “strong explanatory power,” yet R² for WE and RI are not reported. Add R² (with CIs) for endogenous constructs. -
OC treated as unidimensional despite mixed item content.
The OC item set mixes affective/normative/continuance content (e.g., identification, loyalty, difficulty leaving). Either justify a second-order OC or show that a single-factor solution fits better than a three-factor alternative; at minimum, discuss implications. -
Procedural bias during data collection.
The author “reviewed the questionnaires to ensure each item was answered,” which may introduce demand characteristics and reduce anonymity. Clarify procedures (e.g., envelopes, drop-boxes) and acknowledge as a limitation. -
Population figure in Introduction needs verification/precision.
“Monthly average number of resident veterans ranged between ~22,000–24,000” across 16 homes appears very high; ensure the statistic truly refers to Veterans Homes residents, not broader veteran services, and cite precisely
-
Model fit indices mix outdated and modern metrics.
Reporting GFI/AGFI and Gamma hat is unusual; keep them if you wish, but ensure SRMR, RMSEA, CFI/TLI remain primary; they look acceptable (RMSEA =.028; SRMR =.067; CFI/TLI =.978/.977). -
Path coefficients and mediation are reported, but controls are absent.
Include theoretically justified covariates (e.g., age, tenure, gender) to test robustness of direct and indirect paths; retention intentions are known to vary with these. Paths currently: OC→WE =.619; WV→WE =.347; WE→RI =.401; OC→RI =.254; WV→RI =.169; indirect effects OC→RI via WE =.248; WV→RI via WE =.139 (all p≤.001).
-
Potential over-estimation due to construct overlap and method bias.
The semantic overlap between WV and WE items plus single-source measurement likely inflates the WV↔WE association and their paths to RI; true effects may be smaller after purifying scales and adjusting for CMB/clustering. (See Items cited above.) -
Misinterpretation: “greater implications for retention.”
The conclusion that OC’s effect “holds greater implications” than WV is not supported without a formal difference test and may be sample-specific; soften language or provide statistical contrasts -
Under-reporting of limitations.
Please expand limitations to include: cross-sectional design (no causality), potential CMB, non-probability sampling and limited generalizability, clustering not modeled, on-site administration/author presence, and construct specification issues
Looking forward reading your revised paper
Best wishes
Round 2
Reviewer 1 Report
Comments and Suggestions for Authors
I appreciate the changes made based on my initial review. I would like to highlight the ethical and deontological issues of the research process that were not clarified in the first version of the article.
Reviewer 3 Report
Comments and Suggestions for Authors
Thank you for addressing the comments